# Botulinum Toxin a Injection Combined with Radial Extracorporeal Shock Wave Therapy in Children with Spastic Cerebral Palsy: Shear Wave Sonoelastographic Findings in the Medial Gastrocnemius Muscle, Preliminary Study

**DOI:** 10.3390/children8111059

**Published:** 2021-11-17

**Authors:** Dong Rak Kwon, Dae Gil Kwon

**Affiliations:** 1Department of Rehabilitation Medicine, Catholic University of Daegu School of Medicine, Daegu 42472, Korea; 2Department of Rehabilitation Medicine, Comprehensive and Integrative Medicine Hospital, Daegu 42473, Korea; cateyesn@naver.com

**Keywords:** cerebral palsy, spasticity, botulinum toxin, extracorporeal shock wave, shear wave ultrasound

## Abstract

Therapeutic strategies to boost the effect of botulinum toxin may lead to some advantages, such as long lasting effects, the injection of lower botulinum toxin dosages, fewer side effects, and lower costs. The aim of this study is to investigate the combined effect of botulinum toxin A (BTA) injection and extracorporeal shock wave therapy (ESWT) for the treatment of spasticity in children with spastic cerebral palsy (CP). Fifteen patients with spastic CP were recruited through a retrospective chart review to clarify what treatment they received. All patients received a BTA injection on gastrocnemius muscle (GCM), and patients in group 1 underwent one ESWT session for the GCM immediately after BTA injection and two consecutive ESWT sessions at weekly intervals. Ankle plantar flexor and the passive range of motion (PROM) of ankle dorsiflexion were measured by a modified Ashworth scale (MAS) before treatment and at 1 and 3 month(s) post-treatment. In group 1, the shear wave velocity (SWV) of GCM was measured. The PROM and MAS in group 1 and 2 before treatment significantly improved at 1 and 3 month(s) after treatment. The change in PROM was significantly different between the two groups at 1 and 3 month(s) after treatment. The SWV before treatment significantly decreased at 1 month and 3 months after treatment in group 1. Our study has shown that the combination of BTA injection and ESWT would be effective at controlling spasticity in children with spastic CP, with sustained improvement at 3 months after treatment.

## 1. Introduction

Cerebral palsy (CP), mostly caused by brain damage before, during, or after birth, is characterised by abnormal development of posture and movement. Spasticity in children with CP may result in various disabilities, including joint contractures, decreased muscle strength, and a reduced ability to perform activities of daily living. A wide range of therapeutic strategies has been applied to manage spasticity, including physical modalities, oral medication, peripheral neuromuscular blockade, intrathecal agents, and surgical procedures [1].

Among those, botulinum toxin A (BTA) injection has been suggested as a powerful and safe treatment method for focal spasticity in both the upper and lower limbs of children with CP [2,3,4,5]. The relaxation of spastic muscles resulting from BTA injection improves the function of upper and lower limbs as well as gait, and prevents fixed contractures [6,7]. BTA works by inhibiting acetylcholine release from nerve endings in spastic muscles [8].

Extracorporeal shockwave therapy (ESWT) also reduces spasticity in adults with stroke or children with spastic CP [9,10,11,12,13,14,15], although the mechanism remains unclear. However, it seems ESWT alters the chronic hypertonic muscles rheologically, and reduces the stiffness of intramuscular connective tissue. [16]. Even with a single application of ESWT, furthermore, blood flow to muscle tissues may increase, and its repeated application may prolong the effect [17]. Additionally, ESWT is a promising physical method for chemical materials, e.g., genes, to be delivered into cells [18]. For instance, one study demonstrated that ESWT had quickened the anaesthetic action of EMLA cream in an experiment using rat caudal nerves, which would be presumably because the shockwave-mediated transdermal drug was able to be delivered during ESWT [19].

Based on this background, we hypothesised that the combination of BTA and ESWT would yield synergetic therapeutic effects, in the reduction of spasticity in CP. To date, only one study of children with CP has looked into the combined effects of BTA and ESWT [20]. However, it was unable to verify the treatment effects in the study due to its limitations as follows: Firstly, the follow-up period was only 1 month; hence, only short-term effects were able to be evaluated. In clinical settings, long-term effects must be evaluated to verify functional improvement. Secondly, muscle stiffness was evaluated using B-mode echogenicity and compression real-time sonoelastography (RTS). B-mode ultrasound can be performed within a short period of time in a clinical setting; however, it has downsides, being only semiquantitative and prone to inter-observer error [21]. Compression RTS is an ultrasound-based technique which enables the real-time evaluation of tissue elasticity. The principle of this technique is based on the premise that tissue compression can generate strain and cause displacement within the tissue, with the strain in soft tissue being greater than that in firm tissue. However, the technique of compression RTS has several limitations: the compression method yields inaccurate results, inter-operator reproducibility is limited, and only semiquantitative evaluation methods are used [21].

To overcome the limitations of compression RTS, an alternate technique called shear-wave sonoelastography (SWS) which utilizes the acoustic radiation force impulse (ARFI) imaging was introduced. ARFI is a quantitative and objective method of imaging for analysing tissue stiffness. In this technique, the degree of tissue displacement is directly associated with the intensity of forces applied, and inversely related to tissue stiffness. Previous studies reported that SWS can detect the minimal change of tissue stiffness in spastic muscle [22,23,24,25,26].

Taking into account the importance of synergistic, effective, and long-lasting spasticity treatment as well as the issues described above, we aimed to explore the combined effect of BTA injection and ESWT to control spasticity in patients with spastic CP using SWS. To the best of our knowledge, the present preliminary study is the first to report such an expedient treatment and evaluation method in patients with spastic CP.

## 2. Materials and Methods

### 2.1. Participants

This was a retrospective study. Twenty-five potential participants were identified in a retrospective chart review, and fifteen of them were selected as research participants. Clinical information was assessed and evaluated. The Institutional Review Board (IRB) and Independent Ethics Committee (IEC) of the University Medical Center approved the study protocol (IRB No. CR-18-046). Because all the participants were 17 years of age or younger, informed consent was obtained from their parents. A total of 15 patients with spastic CP were evaluated based on specific inclusion/exclusion criteria for injections into tender points of the gastrocnemius muscle (GCM) identified through a chart review. The inclusion criteria were as follows: (1) definite diagnosis of spastic CP by a physiatrist specialising in paediatric rehabilitation, (2) able to independently ambulate with or without walking aid, (3) spasticity while walking with ankle equinus, and (4) dynamic contracture in the ankle. Dynamic ankle contracture was determined in cases where ankle equinus was observed during gait. Passive ankle dorsiflexion was measured with the knee extension. The exclusion criteria were as follows: (1) >13 years or <2 years of age, (2) previous BTA injection in the GCM in the past 6 months, (3) fixed contracture in the ankle, and (4) experience of surgery on the lower limbs or serial casting of the ankle within 12 months.

Through the retrospective chart review, we assessed which kind of treatment the children received. The children were divided into two groups according to the treatment they received. Randomization was not performed because it was a retrospective study.

### 2.2. Intervention

Children in groups 1 and 2 received a BTA (Meditoxin; Medytox Inc., Seoul, Korea) injection at the gastrocnemius (GCM) muscle under sonographic guidance (Figure 1A). The dose of BTA for hemiplegic CP was 4 U/kg and that for diplegic CP was 6 U/kg. Children in group 1 received the combination treatment with one ESWT session (BTL-5000-unit radial type, Columbia, USA; using the following parameters: energy intensity, 0.06 mJ/mm^2^; total shot dose, 1500 shocks; and frequency, 4 Hz) for the GCM immediately after BTA injection and 2 consecutive ESWT sessions at weekly intervals (Figure 1B). Children in group 2 did not receive ESWT. All children received the same outpatient rehabilitation program during and after combined treatment. In all of the children, the frequency for outpatient rehabilitation treatment was undertaken twice a week for three months. The BTA injection and ESWT was performed by one physiatrist. The intervention group was not blinded because this was a retrospective study, and sham ESWT was not available. 

### 2.3. Clinical Assessment

Clinical assessments, including of the spasticity of the ankle plantar flexor and passive range of motion (PROM) of ankle dorsiflexion, were performed before the combined treatment and at 1 and 3 months after the treatment. The PROM of ankle dorsiflexion was measured in the supine position while extending the knee using a two-arm goniometer. The spasticity of the ankle plantar flexor was measured using the modified Ashworth scale (MAS), which is a 6-point scale that rates limb resistance when relaxed to rapid passive stretch (zero point refers to muscle tones that are not increased) [27,28].

### 2.4. Shear-Wave Sonoelastography

Shear-wave velocity (SWV, measured in m/s) of the medial GCM was measured using SWS. A physiatrist with 12 years’ experience in ultrasound and 7 years in sonoelastography implemented B-mode ultrasound and SWS using an ultrasound system commercially available in the 4–9 MHz frequency range with a linear probe (Siemens ACUSON S2000; Siemens Healthcare, Erlangen, Germany). B-mode ultrasound and SWS were performed on the longitudinal view of the medial GCM, and only minimal compression was applied with the transducer weight. While being scanned, the participants were asked to lie prone with their feet sticking out from the examination table. Scanning was discontinued when reflexive or voluntary contraction occurred in the lower limb muscles. Both ends of the tendons, the musculotendinous junction, and the belly of the medial GCM were sequentially identified by scanning the legs from proximally to distally, and the medial GCM was clearly distinguished from the other muscles on ultrasound (Figure 2A). SWS was performed by the same physiatrist in the whole study.

Quality mapping software is utilised due to its usefulness in determining whether an adequate shear wave has occurred [29]. Before measuring SWV, a quality map image was automatically generated by the software (Figure 2B). Subsequently, SWS was measured at a fixed point on the medial GCM, i.e., midway between two reference points, A and B, where point A was located one-third along a longitudinal line from the midpoint between the medial and lateral malleoli to the midpoint between the medial and lateral epicondyles, and point B was located at the medial end of a transversal line that was perpendicular to that longitudinal line (Figure 2C).

### 2.5. Statistical Analysis

Statistical analysis was performed using SPSS version 22.0 (SPSS, Chicago, IL, USA), with the level of significance set at <0.05. Changes in the PROM of ankle dorsiflexion, SWV of the GCM were assessed using repeated-measures analysis of variance (RM-ANOVA) from baseline to 1 and 3 month(s) after BTA injection. The MAS of ankle plantar flexors was assessed with the Friedman test from baseline to 1 and 3 months after BTA injection. Comparison between group 1 and 2 was performed by a Mann–Whitney U test. Post hoc power analysis was performed and the value was >95.

## 3. Results

Of the 25 patients with spastic CP identified through retrospective review, only fifteen were selected as our study participants (Figure 3) The participants were divided into group 1 (seven patients; four boys and three girls; seven right legs, six left legs; mean age: 76.46 ± 24.69 months; and mean body weight: 20.16 ± 8.70 kg) and group 2 (eight patients; four boys and four girls; seven right legs and seven left legs, mean age 109.2 ± 20.4 months; and mean body weight 27.5 ± 10.3 kg). The level of their mean gross motor function classification system (GMFCS) was 3.2 ± 1.2 in group 1 and 2.9 ± 0.8 in group 2.

The mean PROM of ankle dorsiflexion while the knee was extended in groups 1 and 2 before treatment (−14.1° and 5.7°, respectively) was significantly increased at 1 month (6.2° and 15.3°, respectively) and 3 months (0.2° and 12.0°, respectively) after treatment (Table 1, *p* < 0.01). The mean MAS score of the ankle plantar flexor in groups 1 and 2 before treatment (3.2 and 3.2, respectively, median: 3 and 3, IQR: 1 and 2, respectively) was significantly decreased at 1 month (1.4 and 2.0, respectively, median: 1 and 2, IQR: 1 and 1, respectively) and 3 months (1.8 and 2.5, respectively, median: 2 and 2, IQR: 1 and 1, respectively) after treatment (*p* < 0.01). The change in the PROM of ankle dorsiflexion was significantly different between the two groups before treatment and at 1 and 3 months after treatment (*p* < 0.01). The change in MAS score was not significantly different between the two groups (*p* > 0.05). The mean SWV of the GCM before treatment (2.7 m/s) significantly decreased at 1 month (1.8 m/s) and 3 months (2.1 m/s) after treatment in group 1 (*p* < 0.01).

## 4. Discussion

The key results of this study revealed that combined BTA injection with ESWT was effective at controlling spasticity in the children with CP, and that the effect was maintained for 3 months or longer. The mean MAS, PROM of dorsiflexion, and SWV significantly decreased at 1 and 3 months after injection. The minimal clinically important difference of MAS was reported as 0.73; therefore, there was clinically significant improvement of MAS in both groups [30]. To our knowledge, prior to our study, the quantification of the treatment effect of combined BTA injection with ESWT using SWS has not been attempted.

BTA injection combined with ESWT had an anti-spastic effect in patients with CP. Importantly, the age of the participants may have affected the outcomes. Spasticity is determined by two major components, i.e., neural and biomechanical components. The neural component confers resistance to passive movement and is a velocity-dependent phenomenon, whereas the biomechanical component involves a high collagen content in the spastic muscle. In the spastic muscles of children with CP, the collagen content increases and muscle stiffness is significantly correlated with the amount of collagen in the muscle [31]. A previous study demonstrated that the biomechanical component of the spastic muscle and the non-reducible collagen content may be much higher in children aged 5 years and older than in ones under 5 years of age [5]. In the present study, most of the participants were 5 years of age and older; therefore, it appears that the ESWT influenced biomechanical components more than neural components.

ESWT-mediated intramuscular drug delivery has another beneficial effect. One previous study indicated that ESWT would accelerate the anaesthetic effects of EMLA cream based on the study result of rat caudal nerves showing shock wave-mediated transdermal drug delivery occurred during ESWT [19]. These results are consistent with our study results, indicating a synergetic therapeutic effect of BTA injection combined with ESWT on spasticity. The blocking effect of BTA injection on the neuronal component of spasticity decreased with time, suggesting that ESWT conferred shockwave-induced poration and had a therapeutic effect for up to 3 months.

Moreover, ESWT contributes to specific cellular metabolic effects, enhancing enzymatic and non-enzymatic nitric oxide (NO) synthesis [32]. NO plays a role in neuromuscular junction formation in the peripheral nervous system, as well as in essential physiological functions of the central nervous system, including neurotransmission, memory, and synaptic plasticity [8,9]. In addition, NO-induced angiogenesis results in muscle and tendon neovascularization, thereby reducing muscle stiffness. Previous studies have reported that inhibition of the stretch reflex may induce ESWT to work [9,16,33].

However, BTA has several drawbacks. In some patients, responses to BTA injections were observed to have stopped over time, presumably because intramuscular connective tissue and fat content in the spastic muscle had increased after BTA injection [34,35,36]. This destruction of the normal muscle structure may result in fibrosis, muscle atrophy and increased muscle stiffness. Furthermore, in children with CP who have severe spasticity, BTA alone is often insufficient [9]. For this reason, other physical (i.e., non-pharmacological) modalities, such as ESWT, have been suggested for the treatment of spasticity.

Application of ESWT to treat spasticity of various aetiologies has been investigated in various studies. An article related to it was first published in 1997, which showed the safety and effectiveness of ESWT applied in the hypertonic muscles of young people [37]. A minimum of 500 impulses would be required to produce cellular stimulation, while 2500 impulses can generate necrosis [38]. Hence, we selected 1500 impulses to be delivered in each of three sessions, referring to the method used in a previous study [11].

Among previous studies, the combination of BTA and ESWT for spasticity treatment has been applied only in two studies. One of them compared ESWT administered after BTA injection to electrical stimulation given after BTA injection for the management of spasticity in patients who had suffered stroke [39]. The results showed that ESWT enhanced the effect of BTA to a greater extent than electrical stimulation, assumably because ESWT and electrical stimulation may act on different sites. Electrical stimulation improved the BTA effect by facilitating diffusion of BTA, while ESWT acts independently of BTA; that is, ESWT acts mechanically by reducing muscle hypertonicity and trophically by conferring neovascular effects on spastic muscles. The other study compared ESWT performed after BTA injection to BTA injection administered alone to treat spasticity in children with CP [20]. That study showed significant differences in MAS score between groups, but not in Tardieu scale. Unlike the Tardieu scale, the MAS score can assess the sum of rheological factors either related or unrelated to the nervous system due to the fact that increased resistance to passive movement depends on both the stretch reflex and increased muscle stiffness [40]. The authors concluded that the synergistic effect of ESWT and BTA would attribute to BTA-induced inhibition of acetylcholine release at the neuromuscular junction combined with rheological and non-neural effects of ESWT on muscle stiffness. In the current study, the MAS score decreased significantly over time, as did the SWV. This corresponds with the results of previous studies.

In the current study, the PROM and MAS score decreased significantly over time. However, the change in MAS score was not significantly different between the two groups. This is because the neural component is greater than the non-neural component in the MAS. MAS is a method to check velocity-dependent resistance. The neural component is evoked by the velocity-dependent phenomenon [4]. The velocity-dependent increase in muscle resistance arises from the chronic loss of supraspinal inhibitory inputs into alpha motor neurons and is accompanied by exaggerated spinal H-reflexes [41]. The neural component can be inhibited by the administration of BTA but not ESWT. Therefore, the MAS score was not different between the two groups.

This was the first study to use SWS to assess the combined treatment effect of BTA and ESWT in children with CP. Conventional compression RTS tends to be operator-dependent and requires a long learning curve. Moreover, it is difficult to reproduce images because the operator would need to apply pressure using the freehand technique, which would lead to unreliable results [4]. Both previous studies involving combined treatment using BTA and ESWT concluded that the treatment would act through the rheological effect of ESWT [20,39]. To justify such a conclusion, an objective tool to evaluate muscle stiffness would be essential. However, one study did not use any imaging modality [39], while the other used only compression RTS [20]. Therefore, the results of both studies were inconclusive and based on weak evidence.

The present study has several limitations. Firstly, its design was retrospective; therefore, we may not have collected comprehensive records of the participants. Secondly, in this pilot study, there were a limited number of subjects; therefore, further validation through large-scale studies would be required. Thirdly, the study was conducted in a single group of patients because most parents of the participants wanted to receive both BTA and ESWT in our clinic. Although similar comparisons have been performed in other studies of BTA and ESWT, it would be more informative if comparing the efficacy and adverse effects of combined treatments in patients with spastic CP to ones in previous studies. Fourthly, the study results cannot be generalised to all patients with CP because the subjects were limited to patients with spastic CP. Fifthly, the age and weight were different between two groups; we cannot eliminate this confounder because our study was a retrospective study, and the sample size was small. Sixthly, the follow-up period after treatment was relatively short. The participants were only evaluated for 3 months; therefore, longer follow-up studies would be needed to determine the natural course after combined treatment of BTA and ESWT in children with spastic CP. Lastly, we did not evaluate the intra- and inter-rater reliability of SWV because most of the participants and their parents did not agree to the study protocol. Fortunately, however, the high reliability of SWV had already been reported in our previous study [21]. In the future, we will conduct another study without these limitations.

## 5. Conclusions

This study showed that BTA injection combined with ESWT may be effective in controlling spasticity in children with spastic CP by demonstrating that clinical and imaging parameters were improved and maintained 3 months after treatment. Therefore, the appropriate combination of BTA injection and ESWT may be an effective treatment to reduce spasticity in children with spastic CP.

## Figures and Tables

**Figure 1 children-08-01059-f001:**
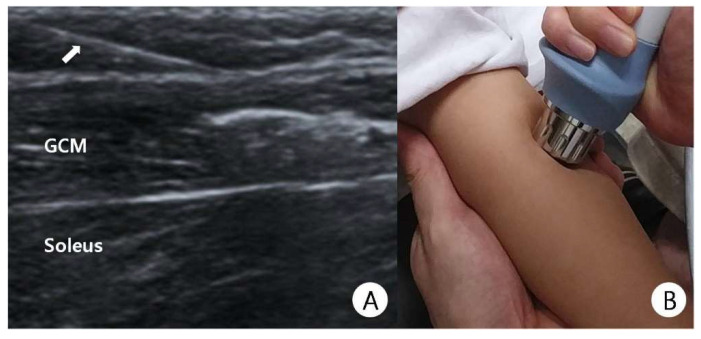
(**A**) Botulinum toxin A injection at the gastrocnemius (GCM) muscle under sonographic guidance (white arrow: needle). (**B**) Application of radial extracorporeal shock wave therapy (ESWT) on the GCM of an 87-month-old child with spastic cerebral palsy.

**Figure 2 children-08-01059-f002:**
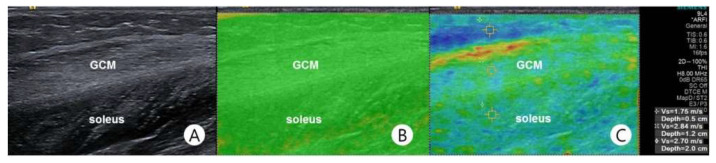
Representative longitudinal acoustic radiation force impulse (ARFI) image of the medial gastrocnemius muscle. (**A**) B-mode ultrasonography. (**B**) “Quality map” indicates the quality and reliability of the shear wave measurements. These correspond to the areas where the shear waves with sufficient quality for quantification are shown in green. (**C**) The shear-wave velocity (2.84 m/s) was measured in the region of interest of the medial gastrocnemius muscle using ARFI imaging.

**Figure 3 children-08-01059-f003:**
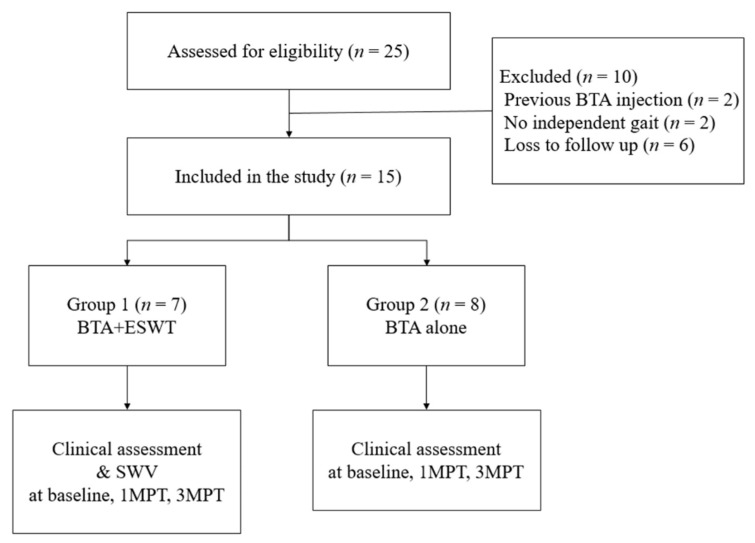
The flowchart of the study.

**Table 1 children-08-01059-t001:** Comparison of changes in range of motion and modified Ashworth scale post injection between group 1 and 2.

	PROM		MAS	
	Group 1	Group 2	Group 1	Group 2
Baseline	−14.1 ± 20.4	5.7 ± 1.5	3.2 ± 0.4	3.2 ± 0.8
1MPT	6.2 ± 7.2 ^(a)^	15.3 ± 4.4 ^(a)^	1.4 ± 0.6 ^(c)^	2.0 ± 0.7 ^(c)^
∆	20.2 ± 17.0 ^(b)^	9.6 ± 4.6 ^(b)^	−1.8 ± 0.3	−1.2 ± 0.6
3MPT	0.2 ± 10.5 ^(a)^	12.0 ± 4.1 ^(a)^	1.8 ± 0.5 ^(c)^	2.5 ± 0.6 ^(c)^
∆	14.1 ± 20.4 ^(b)^	6.2 ± 3.6 ^(b)^	−1.4 ± 0.7	−0.8 ± 0.4

Values are presented as mean ± standard deviation. Group 1: BTA with ESWT, Group 2: BTA without ESWT, BTA, botulinum toxin injection, EWST, extracorporeal shock wave therapy, MPT, month post-treatment time; PROM, passive range of motion; MAS, Modified Ashworth Scale; Δ, difference in the mean PROM and MAS between baseline and at each month post-treatment. ^(a)^
*p* < 0.01, derived from repeated measures of ANOVA for assessment time. ^(b)^
*p* < 0.01, derived from Mann–Whitney U test between group 1 and 2. ^(c)^
*p* < 0.01, derived from Friedman test for assessment time.

## Data Availability

Datasets are available on request.

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
