# Peer review of "Botulinum Toxin a Injection Combined with Radial Extracorporeal Shock Wave Therapy in Children with Spastic Cerebral Palsy: Shear Wave Sonoelastographic Findings in the Medial Gastrocnemius Muscle, Preliminary Study"

_children, 2021, doi:10.3390/children8111059_

Round 1
Reviewer 1 Report
I appreciate the attempt of the authors to address the need for better and new treatments for CP related spasticity in infants and children.
Following edits and changes are recommended.
Abstract- Please rewrite the abstract. Add couple of lines about background – indicating why improvement in therapies for spasticity are needed.
The abstract gives an impression of this being a prospective study, when author clearly mentioned in methods section that it is not. Please rewrite to say that explicitly to mention that patient was recruited after retrospective chart review and then consented for prospective intervention
Methods –
Participants section: I think you need to rewrite this section to clarify again the same confusion. Is this a completely retrospective study or recruitment of patient with retrospective part and then consenting for prospective intervention. Authors need to explain how did they distribute patients into two groups ? randomization or what method? This needs to be clarified.
I think otherwise the Participant section is well written.
Intervention section – Please mention who all were blinded to the intervention group. Or no one was. I would suggest a figure explained methodology in general would be better.
Clinical assessment section: I don’t think writers need to define what each point of MAS test stands for.
Results:
The data on age groups makes me suspect that these two groups are not alike to begin with.
Group 2 seems to be older than group 1 with mean age of >100 months vs 72 months. I would think that would have a greater impact on the results than any treatment. Even though gross motor function classification seems similar, authors need acknowledge that is a big confounder.
What about other demographics of these patients?
Authors also need to mention what kind of therapy were these patients obtaining between the two groups, frequency.
Any medications that these patients were taking for muscle relaxation.
I think Authors need work on making sure there are no other reasons for these results before the conclusion.
If authors do not have these data than they should clearly mention them as weakness of the study.
I think there is no need to repeatedly mention table number after each statistic in paragraph 2 of results section.
Discussion: I think discussion needs also mention long term side effects of muscle atrophy with botox injection. FDA has not approved BOTOX for use in children for muscle spasticity reasons. I would want to know to what age would this therapy be applicable? age 4 and above ?
Author Response
I appreciate the attempt of the authors to address the need for better and new treatments for CP related spasticity in infants and children.
Following edits and changes are recommended.
Q1. Abstract- Please rewrite the abstract. Add couple of lines about background – indicating why improvement in therapies for spasticity are needed.
A1. Thank you for your comment. We added the following sentences about our research background.
“Therapeutic strategies to boost the effect of botulinum toxin may lead to some advantages, such as long lasting effects, the injection of lower botulinum toxin dosages, fewer side effects and lower costs..”
Q2. The abstract gives an impression of this being a prospective study, when author clearly mentioned in methods section that it is not. Please rewrite to say that explicitly to mention that patient was recruited after retrospective chart review and then consented for prospective intervention
A2. Thank you for your comment. We added the phrase about retrospective chart review as follows: Fifteen patients with spastic CP were recruited through a retrospective chart review to clarify what treatment they received.
Methods –
Q3. Participants section: I think you need to rewrite this section to clarify again the same confusion. Is this a completely retrospective study or recruitment of patient with retrospective part and then consenting for prospective intervention. Authors need to explain how did they distribute patients into two groups ? randomization or what method? This needs to be clarified.
A3. We are sorry for the confusion. Our research is a retrospective study. We corrected the sentence to clarify as follows:
“Through a retrospective chart review, we assessed which kind of treatment the children received. The children were divided into two groups according to the treatment they received. Randomization was not performed because it was a retrospective study.”
I think otherwise the Participant section is well written.
Q4. Intervention section – Please mention who all were blinded to the intervention group. Or no one was. I would suggest a figure explained methodology in general would be better.
A4. Thank you for your comment. It was not blinded because this was a retrospective study, and sham ESWT was not available. We added new sentences, and also added a new figure for methodology which is Figure 3.
Q5. Clinical assessment section: I don’t think writers need to define what each point of MAS test stands for.
A5. We also agree with your comment. We deleted the explanation for MAS test.
Results:
Q6. The data on age groups makes me suspect that these two groups are not alike to begin with.
Group 2 seems to be older than group 1 with mean age of >100 months vs 72 months. I would think that would have a greater impact on the results than any treatment. Even though gross motor function classification seems similar, authors need acknowledge that is a big confounder.
What about other demographics of these patients?
A6. Thank you for your comment. We strongly agree with your opinion. We added other demographics, sex, and weight at the result section. Also, we added a sentence about limitations in the discussion section.
Q7. Authors also need to mention what kind of therapy were these patients obtaining between the two groups, frequency.
Any medications that these patients were taking for muscle relaxation.
I think Authors need work on making sure there are no other reasons for these results before the conclusion.
A7. All the patients neither take any medications nor receive other treatments except a rehabilitation program, ESWT, and BTA injection. We added the sentence about frequency of rehabilitation program.
Q8. If authors do not have these data than they should clearly mention them as weakness of the study.
A8. Thank you for your comment. We added a sentence about the weakness of the study.
Q9. I think there is no need to repeatedly mention table number after each statistic in paragraph 2 of results section.
A9. Thank you for your comment. We deleted the repeated table number except the first mention.
Q10. Discussion: I think discussion needs also mention long term side effects of muscle atrophy with botox injection. FDA has not approved BOTOX for use in children for muscle spasticity reasons. I would want to know to what age would this therapy be applicable? age 4 and above ?
A10. Thank you for your comment. We added a phrase about muscle atrophy by BOTOX in the discussion section. In our country, BOTOX was approved for children over 2 years old.

Reviewer 2 Report
Dear authors:
The aim of this study is to investigate combined effect of botulinum toxin A (BTA) injection and extracorporeal shock wave therapy (ESWT) for treatment of spasticity in children with spastic cerebral palsy (CP). Fifteen patients with spastic CP were recruited. All patients received BTA injection on gastrocnemius muscle (GCM), and patients in group 1 underwent one ESWT session for the GCM immediately after BTA injection and two consecutive ESWT sessions at weekly intervals. Modified Ashworth Scale (MAS) of ankle plantar flexor and passive range of motion (PROM) of ankle dorsiflexion were measured before treatment and at 1 and 3 month(s) post-treatment. In group 1, the shear wave velocity (SWV) of GCM was measured. The PROM and MAS in group 1 and 2 before treatment significantly improved at 1 and 3 month(s) after treatment. The change in PROM was significantly different between the two groups at 1 and 3 month(s) after treatment. The SWV before treatment significantly decreased at 1 month and 3 months after treatment in group 1. Our study have shown that combination of BTA injection and ESWT would be effective at controlling spasticity in children with spastic CP, with sustained improvement at 3 months after treatment.
The study is the great interest and it is well written,
I suggest some minor changes:
- A flowchart with the participants in group 1 and group 2, although it was a retrospective study.
- Future lines to research after limitations
Author Response
Reviewer 2
The study is the great interest and it is well written,
I suggest some minor changes:
Q1. A flowchart with the participants in group 1 and group 2, although it was a retrospective study.
A1. Thank you for your comment. We added a flowchart which is Figure 3.
Q2. Future lines to research after limitations
A2. Thank you for your comment. We added a sentence about our future study in the discussion section.
